# Mercury Exposure and Poor Nutritional Status Reduce Response to Six Expanded Program on Immunization Vaccines in Children: An Observational Cohort Study of Communities Affected by Gold Mining in the Peruvian Amazon

**DOI:** 10.3390/ijerph16040638

**Published:** 2019-02-21

**Authors:** Lauren Wyatt, Sallie Robey Permar, Ernesto Ortiz, Axel Berky, Christopher W. Woods, Genevieve Fouda Amouou, Hannah Itell, Heileen Hsu-Kim, William Pan

**Affiliations:** 1Nicholas School of the Environment, Duke University, Durham, NC 27710, USA; lauren.wyatt@alumni.duke.edu (L.W.); axel.berky@duke.edu (A.B.); 2Duke Human Vaccine Institute, Duke University Medical Center, Durham, NC 27710, USA; sallie.permar@duke.edu (S.R.P.); genevieve.fouda@duke.edu (G.F.A.); haitell@uw.edu (H.I.); 3Global Health Institute, Duke University, Durham, NC 27710, USA; ernesto.ortiz@duke.edu; 4Department of Medicine, Duke University School of Medicine, Durham, NC 27710, USA; chris.woods@duke.edu; 5Department of Civil and Environmental Engineering, Duke University, Durham, NC 27710, USA; hsukim@duke.edu

**Keywords:** mercury, exposure, immune response, nutritional status, ASGM, Madre de Dios, Peruvian Amazon

## Abstract

*Background:* Poor nutritional status combined with mercury exposure can generate adverse child health outcomes. Diet is a mediator of mercury exposure and evidence suggests that nutritional status modifies aspects of mercury toxicity. However, health impacts beyond the nervous system are poorly understood. This study evaluates antibody responses to six vaccines from the expanded program on immunization (EPI), including hepatitis B, *Haemophilus influenzae* type B, measles, pertussis, tetanus, and diphtheria in children with variable hair mercury and malnutrition indicators. *Methods:* An observational cohort study (*n* = 98) was conducted in native and non-native communities in Madre de Dios, Peru, a region with elevated mercury exposure from artisanal and small-scale gold mining. Adaptive immune responses in young (3–48 months) and older children (4–8 year olds) were evaluated by vaccine type (live attenuated, protein subunits, toxoids) to account for differences in response by antigen, and measured by total IgG concentration and antibody (IgG) concentrations of each EPI vaccine. Mercury was measured from hair samples and malnutrition determined using anthropometry and hemoglobin levels in blood. Generalized linear mixed models were used to evaluate associations with each antibody type. *Results:* Changes in child antibodies and protection levels were associated with malnutrition indicators, mercury exposure, and their interaction. Malnutrition was associated with decreased measles and diphtheria-specific IgG. A one-unit decrease in hemoglobin was associated with a 0.17 IU/mL (95% CI: 0.04–0.30) decline in measles-specific IgG in younger children and 2.56 (95% CI: 1.01–6.25) higher odds of being unprotected against diphtheria in older children. Associations between mercury exposure and immune responses were also dependent on child age. In younger children, one-unit increase in log_10_ child hair mercury content was associated with 0.68 IU/mL (95% CI: 0.18–1.17) higher pertussis and 0.79 IU/mL (95% CI: 0.18–1.70) higher diphtheria-specific IgG levels. In older children, child hair mercury content exceeding 1.2 µg/g was associated with 73.7 higher odds (95% CI: 2.7–1984.3) of being a non-responder against measles and hair mercury content exceeding 2.0 µg/g with 0.32 IU/mL (95% CI: 0.10–0.69) lower measles-specific antibodies. Log_10_ hair mercury significantly interacted with weight-for-height z-score, indicating a multiplicative effect of higher mercury and lower nutrition on measles response. Specifically, among older children with poor nutrition (WHZ = −1), log_10_ measles antibody is reduced from 1.40 to 0.43 for low (<1.2 µg/g) vs. high mercury exposure, whereas for children with good nutritional status (WHZ = 1), log_10_ measles antibody is minimally changed for low vs. high mercury exposure (0.72 vs. 0.81, respectively). *Conclusions:* Child immune response to EPI vaccines may be attenuated in regions with elevated mercury exposure risk and exacerbated by concurrent malnutrition.

## 1. Introduction

Vaccination is an important public health tool for preventing morbidity and mortality worldwide. Vaccine response is dependent on many factors, including environmental pollutant exposures, which have become an increasingly important risk factor. Some exposures like polychlorinated biphenyls have strong associations with reduced responses that are observed in multiple populations [1,2], while for others, including heavy metals, associations have only been observed under certain conditions, such as malnutrition. Health impacts associated with environmental pollutant exposure may be overlooked due to these relationships being confounded by factors such as malnutrition that have the potential to interact with the exposure [3,4]. These complex interactions can occur through a toxicant influencing nutrient metabolism, nutritional status affecting toxicant absorption or excretion, or both influencing the same endpoint [5]. This paper focuses on mercury exposure, where diet plays a key role in the connection between exposure and health outcomes. In Amazonian populations, fish consumption is the primary exposure route [6,7,8], though consumption of some dietary items is associated with reduced exposure [9,10]. In Peru, exposure through vaccinations via thiomersal is thought to be minimal, as only the flu vaccine contains this compound, similar to the US. Mercury compounds alter a number of immune endpoints that increase disease susceptibility in animal studies. Dietary methylmercury exposure suppresses primary immune responses such as cell number and cell proliferation and also reduces general and specific immunoglobulin antibody levels [11,12,13]. Mercury exposure is also hypothesized to increase the risk of nutrient deficiencies, such as anemia due to impaired hemoglobin function from mercury competing with iron for binding sites [14]. In our study region, elevated mercury content in hair was associated with decreased hemoglobin levels in children under 12 years old [15]. Similar observations of increased anemia and anemia severity in children occurs with elevated lead exposure [16,17,18,19]. 

Nutritional status has an important but complex impact on immune function as malnutrition can increase susceptibility to infection, but infections can also aggravate malnutrition by exacerbating nutrient loss. Severe malnutrition is linked to a suppressed adaptive immune response to routine vaccination against tetanus [20], hepatitis B [21,22], typhoid [23], and measles [24]. Nutritional status may also contribute to the contrasting results reported between mercury exposure and adaptive immune outcomes in children. In the Faroe Islands, children exposed to organic mercury had a slight but not significant negative correlation between mercury concentration and diphtheria and tetanus antibody levels [25], while two National Health and Nutrition Examination Survey (NHANES) studies observed that elevated mercury exposure was significantly associated with decreased levels of measles and rubella antibodies for a subgroup of male children with low vitamin B12 [26,27]. While nutrition is one possible explanation for the contrasting observations in these studies, vaccine type is another. Toxoid vaccines were examined in the Faroe Islands study and live attenuated virus vaccines in the NHANES studies. The efficacy of live attenuated vaccines may be attenuated as methylmercury has been observed to alter viral replication [28] and impair viral clearance [29] in laboratory studies. Impairments in viral clearance could be further exacerbated under conditions of increased immune susceptibility, like undernutrition. Other vaccine types, including protein subunit vaccines, may also be affected by mercury exposure as women with chronic hepatitis B infection were observed to have elevated total blood mercury in an NHANES study [30]. This relationship should be further investigated, taking into consideration both vaccine type and nutritional status.

The goal of this study is to evaluate the influence of mercury exposure, nutritional status, and their interactions with immune responses in children. Data are from an observational cohort study in Madre de Dios (MDD), Peru, a gold-mining region where the majority of the population have hair mercury content exceeding a level associated with impaired child development (1.2 µg/g) [31,32,33]. We test the hypothesis that elevated mercury exposure and concurrent malnutrition attenuate immune response as measured by antibody titers to six expanded program on immunization (EPI) vaccines: measles, hepatitis B (Hep B), *Haemophilus influenzae* type B (Hib), pertussis, tetanus, and diphtheria. Results are interpreted according to vaccine type (live attenuated, subunits, and toxoids) to infer potential pathways through which mercury may mediate immune response. 

## 2. Materials and Methods 

### 2.1. Study Background and Population

This study was conducted in communities adjacent to the Amarakaeri Communal Reserve (ACR) in the Madre de Dios (MDD) and Cusco regions (Figure 1). MDD has a population of ~137,000 and the highest rate of population growth in the country (2.5% per year) [34]. Regional artisanal and small-scale gold-mining (ASGM) has increased substantially over the past decades [35,36,37] and is associated with increased concentrations of mercury in the environment [38]. 

Elemental mercury is used as an amalgamating agent in the mining process and is released into the atmosphere when the gold-mercury amalgam is heated. Improper handling and atmospheric deposition result in mercury entering terrestrial and aquatic environments where it can be converted to methylmercury and bioaccumulate in fish populations, the primary route of mercury exposure in this region [10]. Mercury exposure in MDD poses a considerable human health risk as diet is the most common exposure route, but inhalation exposure is also possible in families located near amalgam processing [10,38].

In addition to elevated mercury exposure, communities in this region have increased poverty and inadequate access to clean water and health services [39]. Rural and native populations are often more disadvantaged, as disparities in resources, infrastructure access, and health exist within localized Amazonian regions [40]. Chronic malnutrition and other nutritional deficiencies are common in children in MDD, with anemia prevalent 44.9% of children under 5, compared to the 26.4% in Peru’s capital, Lima [41,42]. Although stunting prevalence estimated from the Demographic Health Survey (DHS) data has reportedly declined in MDD [43], 17.5% of children under 8 in this study were stunted, including 42.9% of children surveyed in native communities. 

### 2.2. Study Design and Data Collection

Children included in this study are a part of a population-based study aimed at evaluating the health status of households in 23 native (indigenous) and non-native communities surrounding the Amarakaeri Communal Reserve (Figure 1). Twelve non-native communities (Boca Colorado, Boca Manu, Caychihue, Choque, Huepethue, Punquiri, Puquiri, Quebrada Nueva, Quimiri, Quincemil, Salvacion, Setapo) and 11 native communities (Boca Isiriwe, Diamante, Isla de los Valles, Masenawa, Palotoa Teparo, Puerto Azul, Puerto Luz, Queros, San Lorenzo, Shintuya, Shipetiari) were included. Two rounds of data collection were conducted: a baseline sampling of all households with WCBA between March and June 2015 (a random sample of 75% of WCBA was drawn in the large urban towns of Salvacion, Huepethue and Quincemil); and a follow-up of WCBA with children between January and April 2016. Children were eligible for this study if they were less than 8 years old at the time of follow-up. The study was designed to detect a 10% decline in antibody titers for children exceeding 2.0 µg/g with 80% power at the 0.05 significance level with data from 140 children. Power calculations were based on prior studies of mercury [26,27] and PCB [1] exposures in children.

Baseline and follow-up data collection included: household surveys to obtain demographic, household, and health information; height (standing or recumbent length for children <24 months) and weight using validated instruments (Omron HBF-514C, Bannockburn, IL, USA); hemoglobin using HemoCue Hb 201+ (Angelholm, Sweden); and hair samples to measure mercury content. Follow-up data also included child vaccination history and blood (serum) samples for vaccine testing. Child vaccination records included the date of administration from the child’s vaccination card and records at the health post. If vaccination information was absent from the card and the health post it was assumed that it did not occur (a conservative assumption to avoid recall bias, but may incorrectly exclude some children from analysis). 204 children under 8 were eligible for this study at follow-up. 43 children did not provide a blood sample and one child’s sample did not coagulate. After omitting children with incomplete vaccination data and missing hair mercury results, a total of 98 samples were available for analysis (Figure 2). 

Parental consent was obtained for all data collection with additional assent from children 6 to 8 years old. Approval to conduct research on human subjects was obtained through the Universidad Peruana Cayetano-Heredia (UPCH) Comité Institucional de Ética (CIE) para Humanos (SIDISI 63056) and the Regional Health Directorate of Madre de Dios (DIRESA-MDD).

### 2.3. Hair Mercury Analysis

Hair samples were obtained during baseline and follow-up visits from the occipital region of the head using stainless steel scissors. Exposure was estimated from the total mercury content in the most proximal 2 cm segment of hair, which corresponds with the estimated exposure over the most recent 2 month period before sample collection [31]. This most proximal 2 cm segment has been observed to be a strong predictor of chronic mercury exposure in MDD [10]. Total mercury was measured by direct combustion, gold amalgamation, atomic absorption spectrometry (Milestone DMA-80, Milestone SRL, Sorisole, Italy). The instrument calibration was verified by analysis of a hair standard reference material (ERM-DB001) once every 10 samples in a batch run. Accepted measurements were within 10% of certified value and the detection limit was 1 ng Hg. 

### 2.4. Antibody Analysis

Child serum samples for antibody analysis were collected either from finger prick or venous blood. Samples were centrifuged as appropriate and stored at −20 °C in the field, then transferred to −80 °C until analysis. Serum antibody concentrations were assessed for total IgG and six routine childhood vaccinations (hepatitis B [Hep B], measles, *Haemophilus influenzae* type B [Hib], diphtheria, pertussis, and tetanus). Samples were run at the Duke Human Vaccine Institute and antibody concentrations were determined using either an enzyme-linked immunosorbent assay (ELISA) or a recently developed pediatric vaccine multiplex assay (PVMA, [44]). ELISA assays were used to measure total IgG and antibodies against Hep B, measles, and Hib. All sample dilutions were the same within each ELISA (total IgG: 1:3,000,000; Hep B: 1:500; measles: 1:100; Hib: 1:25). For each assay, the upper and lower optical density (OD) limits of detection were determined. Samples were repeated if their OD or concentration coefficient of variance was greater than 20%. PVMA was used to measure antibody responses against diphtheria, pertussis, and tetanus with sample dilutions of 1:100. When available, WHO standard (NISBC) were used to calculate the antigen-specific antibody concentration. The concentration of many of the antibodies was determined in “international units” per milliliter (IU/mL) which is an arbitrary quantification that is used for many international standards. The standards in these assays that use IU/mL are: Anti-HepB Surface Antigen Immunoglobulin (WHO 2nd International Standard), Pertussis Antiserum (WHO 1st International Standard), Diphtheria Antitoxin (WHO 1st International Standard), Tetanus Immunoglobulin (WHO 1st International Standard), and Measles IgG. For each antigen, upper and lower detection limits were determined. For all assays, values that fell below the lower detection limit were handled using the midpoint between 0 and the lower detection limit, and values that were above the detection limit were handled by using the upper detection limit. Less than 10% of values were outside of the detection limits for measles, pertussis, tetanus, and diphtheria. 21% of the values were below the detection limit for Hep B and 11% of the values were above the detection limit for Hib.

### 2.5. Statistical Analysis

Community (native vs. non-native) demographics, hair mercury content, anthropometry, hemoglobin (Hb), anemia (Hb < 11.0 g/dL [45]), antibody concentration, and vaccination coverage were compared using geometric means for hair mercury and antibody concentrations, and arithmetic means for others. Anthropometric measurements (height, weight) were converted to height-for-age (HAZ), weight-for-age (WAZ), and weight-for-height (WHZ) z-scores using the R package zscorer (CRAN repository), which calculates z-scores based on WHO child growth standards [46]. Stunting, underweight, and wasting were defined by a z-score (HAZ, WAZ, and WHZ, respectively) less than −2 SD away from the mean. Children with HAZ, WAZ, and WHZ values more than 5 SD away from the mean were excluded from the analysis (*n* = 5 children). Hair mercury content was compared to reference levels calculated from the United States Environmental Protection Agency (USEPA) benchmark dose for a maternal exposure level related to child developmental impairment (1.2 µg/g) and the level recognized by Peru’s government (2.0 µg/g), which is calculated from the WHO provisional intake of methylmercury [31,32]. To describe nutritional status in reference to a healthy population, hemoglobin concentrations, weight, and height measurements were compared to NHANES children that were also <8 years.

The impact of mercury and nutritional factors on vaccine-elicited antibodies was assessed individually and by vaccine type in both a continuous and categorical (protected vs. not protected, responder vs. non-responder) manner. Three vaccine types were included in this assessment: live attenuated (measles), subunit vaccines (Hep B, Hib), and toxoid vaccines (pertussis, diphtheria, tetanus). Protection was determined by antibody concentrations noted to afford protection in the literature. Protection thresholds have been defined for Hep B (10 IU/L), Hib (1.0 µg/mL), diphtheria (0.1 IU/mL), and tetanus (0.1 IU/mL) [47,48,49,50]. For the diseases without previously defined thresholds for protection, classifications of being a responder or non-responder were created from assay specific thresholds. For measles, responders were identified using reference serum to estimate the protection level (1000 IU/mL) and for pertussis, responders had median fluorescence intensity (MFI) ≥ 100 (1.37 IU/mL) as below 100 MFI non-specific binding has been observed to occur [51,52]. The distribution of hair mercury content and antibody concentrations were right skewed; thus, log-transformations (log_10_) were used in subsequent analyses.

#### Risk Factors Associated with Changes in Antibody Concentrations and Levels That Are Indicative of Being Not Protected or a Non-Responder

Generalized linear models (GLMs) were used to evaluate factors associated with: (1) a child’s antibody concentration and (2) a child having antibody levels that would classify them as being not protected or a non-responder for each vaccine antibody. Key variables of interest include: community group (native vs. non-native); child hair mercury content, evaluated as continuous and categorical (exceed or not exceed 1.2 or 2.0 µg/g); and nutritional status (HAZ, WAZ, WHZ, stunting, wasting, underweight, hemoglobin, anemic), considered as continuous and categorical. Additional factors of interest included: variables hypothesized to be correlated with child mercury exposure (i.e., average parenteral hair mercury).

Separate GLMs were fit for young (<4 years) and older (4 to <8 years) children to differentiate stage of vaccination (incomplete / complete) based on Peru’s schedule (Figure 3). As stated above, children were analyzed only if they had a confirmed vaccination history, received at least one dose of a specific vaccine, and at least 20 days had passed since the last vaccination. All models were adjusted for sex and time since last vaccination (age for total IgG model). A community random effect was included for all models to adjust for correlated exposures within communities. Statistics were calculated using R Version 3.4.0 (R Foundation for Statistical Computing, Vienna, Austria) [53].

## 3. Results

### 3.1. Population Characteristics

A total of 161 children were enrolled in this study, 42% were less than 4 years old and 52% were female. Ages of children ranged 3 months to 8 years. Enrolled children had elevated hair mercury content, anemia prevalence, and instances of other malnutrition indices compared to a US population (Table 1). These attributes varied by community type, with the greatest hair mercury content and highest instances of anemia and malnutrition occurring in native communities.

#### 3.1.1. Child and Parental Hair Mercury Content

Overall, 41% of children had a hair mercury level that exceeded the USEPA threshold at baseline, 57% at follow-up (Table 1). Child hair mercury content was significantly higher in native vs. non-native communities at both baseline (2.5 vs. 0.6 µg/g, *p* < 0.001) and follow-up (3.8 vs. 1.0 µg/g, *p* < 0.001). Children in native communities were more likely to have hair mercury that exceeded both the USEPA and WHO levels (above 2.0 µg/g) than children in non-native communities (69% vs. 14% at baseline, 83% vs. 24% at follow-up, Table 1). 

Similar to children, 52% of mothers had a hair mercury level that exceeded the USEPA level at baseline, 54% at follow-up. Average parental hair mercury in native communities was significantly higher than parental hair mercury in non-native communities (4.2 vs. 1.4 µg/g respectively at baseline and 5.1 vs. 1.6 µg/g respectively at follow-up). Parental hair mercury was correlated with child mercury level, with baseline paternal mercury having a correlation of 0.74 with their children at baseline and 0.86 at follow-up. 

#### 3.1.2. Nutritional Status

Anemia and stunting were present in this population, with a higher prevalence in native communities. 43% of children at baseline and 35% at follow-up were anemic (<11.0 g/dL hemoglobin). Native communities had a significantly higher proportion of anemic children compared to non-native communities (60% vs. 37% at baseline and 52% vs. 28% at follow-up, Table 1). Stunted growth in children at baseline (14.9%) was significantly more common in native vs. non-native communities (30% vs. 8.5%), and more common in young vs. older children (17% vs. 6%). Only 4% of children were classified as underweight and 2% wasted (Table 1).

#### 3.1.3. Vaccination Coverage and Response

Vaccination information from health posts or vaccination cards was available for three quarters of children (15% health post and card, 21% card only, and 38% health post only). For younger children (<4 years), the proportion up-to-date with Peru’s vaccination schedule was 74% for Hepatitis B given at birth, 93% for pentavalent, 77% for DPT (diphtheria, pertussis, tetanus), and 37% for MMR (measles, mumps, rubella). For older children (4 to <8 years), the proportion up-to-date was 65% for Hepatitis B given at birth, 84% for pentavalent, 55% for DPT, and 51% for MMR. For all vaccines, vaccination coverage was similar between native and non-native communities. Overall, the percentage of children receiving at least one dose for protection against each antibody was high: 63% for measles, 71% for Hep B, 69% for Hib, 74% for diphtheria, 74% for pertussis, and 74% for tetanus. Among children confirmed to have received at least one dose, these percentages were similar between native and non-native communities (Table 1). 

Antibody levels were similar between community types for total IgG, hepatitis B, diphtheria, pertussis, and tetanus; however, measles, Hib, and diphtheria-specific IgG were lower in native vs. non-native communities (Table 1). Of children that received at least one dose for a particular vaccine, the percent of children without an antibody concentration indicative of protection or with undetectable levels (non-responding) was: 0% against hepatitis B; 38% against measles; 18% against Hib; 35% against diphtheria; 14% against pertussis; and 10% against tetanus. 

### 3.2. Risk Factors Associated with Changes in Antibody Concentrations

The final models to estimate log_10_ antibody concentrations (GLM-Linear) and odds of not being protected or not being a responder against a particular antibody (GLM-Logit) included a hair mercury variable (log_10_ and exceeding EPA or WHO level), a nutritional status variable, the interaction between the mercury and nutritional status variables, sex, and time (time since last vaccination for vaccination antibodies and age for total IgG). 

For GLM-Logit models, only results from models with older children (4 to <8 years) and for measles, Hib, diphtheria, pertussis, and tetanus-specific IgG are shown because no child had antibody levels indicating non-response for Hep B and few younger children had antibody concentrations that could be classified as non-responders for multiple vaccines. The GLM-Linear model included the same co-variates as the GLM-Logit models. Relationships between vaccine response and community type (non-native vs. native), malnutrition, and hair Hg are discussed below. 

The most important findings from these models were that: (1) Hib-specific antibodies were reduced in older children belonging to a native community; (2) lower nutritional status was associated with reduced measles, diphtheria, and tetanus-specific antibodies in younger children; (3) elevated hair mercury content was associated with reduced measles-specific antibodies in all children; and 4) the interaction between elevated hair mercury content and lower nutritional status was correlated with decreased measles and pertussis-specific antibodies and increased tetanus-specific antibodies.

#### 3.2.1. Community Effects

The inclusion of community type in models had little impact on antibody concentrations or odds of not being protected or not being a responder against a particular antibody, except for Hib. In older child models, community type (native vs. non-native) was observed to be a significant predictor of Hib-specific antibodies. In the fully adjusted models, belonging to a native community was associated with decreased log_10_ Hib-specific antibodies (β mean: 0.50 µg/mL, range: 0.40–0.60 µg/mL) and increased odds of not being protected (OR mean: 2.6, range 1.5–12.0) (Appendix A).

#### 3.2.2. Malnutrition

Low hemoglobin was associated with decreased total IgG, measles, diphtheria, and tetanus-specific IgG. In young children, GLM-Logit results demonstrate that a one-unit decrease in hemoglobin was associated with 0.17 lower log_10_ measles (95% CI: 0.04–0.30, *p* = 0.02) and 0.18 increased log_10_ diphtheria-specific (95% CI: 0.01–0.35, *p* = 0.04) IgG levels. Also in young children, a one-unit decrease in HAZ score was associated with 0.19 decreased log_10_ diphtheria-specific IgG levels (95% CI: 0.04–-0.34, *p* = 0.02). log_10_ tetanus-specific IgG levels were additionally reduced by a one-unit increase in WAZ (β = 0.20, 95% CI: 0.03–0.38, *p* = 0.03) and WHZ (β = 0.33, 95% CI: 0.08–0.58, *p* = 0.02) (Figure 4, Appendix A). 

Fewer associations were observed in older children (4 to <8 years old); however, low hemoglobin and low WAZ were associated with decreased diphtheria and Hib-specific antibody levels. A one-unit decrease in hemoglobin was associated with increased odds of being unprotected against diphtheria (OR = 2.56, 95%CI: 1.01–6.25, *p* = 0.04). Additionally, a one-unit decrease in WAZ was associated with 0.54 decreased log_10_ Hib-specific antibodies (95% CI: 0.10–0.98, *p* = 0.02) and increased odds of being unprotected against Hib (OR = 1.79 × 10^6^, 95% CI: 1.05–3.06 × 10^12^, *p* = 0.04) (Figure 4, Appendix A).

#### 3.2.3. Hair Mercury

Elevated mercury exposure was associated with increased pertussis and diphtheria-specific IgG levels in younger children. A one-unit increase in log_10_ child hair mercury content was associated with 0.79 increased log_10_ diphtheria (95% CI: 0.18–1.70, *p* = 0.02), 0.68 increased log_10_ pertussis-specific antibody levels (95% CI: 0.18–1.17, *p* = 0.01) (Figure 4). Replacing child mercury level with parental Hg level resulted in similar relationships identified for pertussis and diphtheria, reflecting the strong child-parent Hg correlation (Appendix A). In older children, elevated mercury exposure was associated with decreased measles-specific antibody levels. Child hair exceeding 1.2 µg/g was associated with 73.70 (95%CI: 2.75–1984.31, *p* = 0.01) increased odds of being a non-responder against measles and when child hair exceeding 2.0 µg/g with 0.32 decreased log_10_ measles-specific antibodies (95% CI: 0.01–0.62, *p* = 0.04) (Figure 4, Appendix A).

#### 3.2.4. Interaction between Mercury and Malnutrition

In younger children, hair mercury levels interacted with HAZ to influence diphtheria response. In combination with a one-unit increase in log_10_ child hair mercury content, log_10_ diphtheria-specific antibodies were reduced 0.35 (95% CI: 0.07–0.65, *p* = 0.02) with a one-unit decrease in HAZ and 1.87 (95% CI: 0.46–3.27, *p* = 0.01) with a stunted condition. The net diphtheria-specific antibodies are estimated to be 7.57 in a child with high hair mercury content (4 µg/g) & good nutrition (HAZ = 1), 1.17 in a child with high hair mercury content and poor nutrition (HAZ = −1), 1.55 in a child with low hair mercury content (1 µg/g) & good nutrition, and 0.65 in a child with low hair mercury content and poor nutrition (Figure 4, Appendix A).

In older children, the interaction of increased mercury content in hair and lower nutritional status was associated with decreased measles and pertussis and increased tetanus-specific antibodies. The interaction between a one-unit decrease in WHZ and child hair exceeding 1.2 µg/g resulted in a 0.23 (95% CI: 0.03–0.53, *p* = 0.03) decrease in log_10_ measles-specific antibodies. The net measles-specific antibodies would be estimated to be 0.81 in a child with high hair mercury content (exceed 1.2 µg/g) and good nutrition (WHZ = 1), 0.43 in a child with high hair mercury content and poor nutrition (WHZ = −1), 0.72 in a child with low mercury content (does not exceed 1.2 µg/g) and good nutrition, and 1.40 in a child with low hair mercury content and poor nutrition. Additionally, the odds of being a non-responder against measles were increased with one-unit decrease in WHZ in combination with child hair mercury content exceeding 1.2 µg/g (OR = 20.49, 95% CI: 1.30–323.76, *p* = 0.03). The relative effect on the odds of being a non-responder against measles were increased 36.42 in a child with high hair mercury content (exceed 1.2 µg/g) and good nutrition (WHZ = 1), 150.05 in a child with high hair mercury content and poor nutrition (WHZ = −1), 10.12 in a child with low mercury content (does not exceed 1.2 µg/g) and good nutrition, and 0.10 in a child with low hair mercury content and poor nutrition (Figure 4, Figure 5, Appendix A). 

With respect to pertussis-specific antibodies, negative interaction terms were observed between a one-unit increase in log_10_ child hair mercury content and an anemic condition (β = 0.78, 95% CI: 0.17–1.39, *p* = 0.01) and child hair mercury exceeded 1.2 µg/g in combination with a stunted condition (β = 0.98, 95% CI: 0.06–1.90, *p* = 0.04). The net pertussis-specific antibodies are estimated to be 0.82 in a child with high hair mercury content (4 µg/g) & good nutrition (not anemic), 3.83 in a child with high hair mercury content and poor nutrition (anemic), 1.37 in a child with low hair mercury content (1 µg/g) & good nutrition, and 0.73 in a child with low hair mercury content and poor nutrition. Log_10_ tetanus antibodies were increased when child hair mercury exceeded 1.2 µg/g in combination with decreased WAZ (β = 0.36, 95% CI: 0.07–0.66, *p* = 0.01) or WHZ (β = 0.28, 95% CI: 0.02–0.54, *p* = 0.03). The net tetanus-specific antibodies are estimated to be 0.47 in a child with high hair mercury content (exceed 1.2 µg/g) and good nutrition (WAZ = 1), 1.18 in a child with high hair mercury content and poor nutrition (WAZ = −1), 1.96 in a child with low mercury content (does not exceed 1.2 µg/g) and good nutrition, and 0.91 in a child with low hair mercury content and poor nutrition (Figure 4, Appendix A).

## 4. Discussion

In this study we report evidence that nutritional deficiencies, environmental exposure to mercury, and their interaction may impair child immune response to specific vaccine antigens among children under 8 years of age in the Peruvian Amazon. Children in this study are from a region in Peru where over 43% of children are anemic, 15% are stunted, and many have hair mercury content that exceeds USEPA and WHO levels (41 and 31%, respectively). These factors contribute to overall poor vaccine response in the region, where only ~60% of children achieve protective antibody levels for measles and diphtheria, and 80–90% achieve protection against Hib, pertussis and tetanus. Our results support the idea that elevated hair mercury and malnutrition pose a dual risk to childhood vaccine response, but the effects vary by age and antibody type. For example, the effect of mercury exposure was in opposing directions for younger and older children; namely, elevated exposure was associated with modest increases in antibodies for children under 4 (diphtheria, pertussis, tetanus, Hib), but decreased antibodies among children 4–8 (pertussis, measles). In contrast, the impact of indicators of malnutrition were generally in similar directions for both younger and older children. In younger children, anemia (<11.0 g/dL) was associated with decreased measles and hepatitis B-specific antibodies, while in older children, lower hemoglobin levels were correlated with decreased tetanus specific antibodies. Similarly, anthropometric measures were associated with lower response to diphtheria, pertussis and tetanus, but mostly in young children.

Vaccination is the most important public health tool for preventing disease in human populations. Based on our data, interactions of concurrent malnutrition and mercury exposure can pose a threat to achieving protective population immunity, particularly for measles response among older children. Risk of interaction effects are likely greater in older children compared to younger due to the immune response impacts of chronic mercury exposure. Living in an ASGM environment can result in long-term exposure and prior research in this region demonstrated that the proximal 2cm hair segment is highly predictive of exposure over the past 12 months [10]. The effect is dramatic: Increasing hair mercury content from 1 to 4 µg/g increased the odds of being a non-responder to measles 36 times and the addition of poor nutritional status (−1 SD WHZ) increased these odds to 150 times. Under high hair mercury content conditions (4 µg/g) poorer nutrition (anemic vs. not anemic) resulted in 1.5× increased pertussis-specific antibodies, while under low hair mercury content conditions (1 µg/g) poorer nutrition resulted in 1.9× decreased antibodies.

Severe and chronic malnutrition are strongly associated with impaired sero-conversion following vaccination [20,21,22]. In this study, we were limited in assessing the impact of chronic malnutrition as few children (<4%) were classified as severely stunted (HAZ < −3). Instead, moderate stunting (HAZ < −2) was used as a proxy to evaluate chronic malnutrition and was not observed to be associated with immunosuppression. Though our measured health indices were underpowered in detecting chronic malnutrition, anemia, which could indicate an acute or chronic health condition, was associated with reduced hepatitis B, measles, and tetanus titers. Anemia was characterized by low hemoglobin concentration and is caused by factors that destroy red blood cells or impair their production, such as nutritional deficiencies (iron, folate) and parasites. In the context of this study, anemia represents a general poor health condition, as the cause of anemia was not determined. Low seroconversion related to hemoglobin level has been observed previously in two sensitive populations, the elderly and hemodialysis patients. In these populations, anemia was associated with a hypo-responsiveness following vaccination to hepatitis B and influenza that was hypothesized be related to nutritional deficiencies [54,55,56,57]. 

Methylmercury’s impact on the immune system includes immunosuppression and enhanced inflammation in laboratory and human studies [27,58,59]. In this study we observed that mercury related impacts not only vary in direction, but are also antigen specific, providing additional evidence of mercury’s complex impact on immune function. Immune suppression was observed though associations with elevated child and paternal hair mercury and reduced measles titers. We also observed some instances of correlations with increased Hib, diphtheria, pertussis, and tetanus titers, which only occurred in the younger age models adjusted for HAZ, which suggests that there could be a stimulatory response to mercury exposure early in life. Early-life exposure is important to aspects of mercury toxicity, including delayed neurodevelopment [60] and immunomodulation [58], and because exposure seasonality has been observed in this region [10], having elevated hair mercury in this timeframe may correspond to a previous sensitive window of time. 

Overall, mercury in combination with lower nutritional status was consistently associated with suppressive impacts on a live attenuated virus vaccine (measles) and two toxoid vaccines (diphtheria and pertussis) and a stimulatory effect on one toxoid vaccine (tetanus). These observations support the hypothesis of altered immune response related to co-morbidities. This has large implications for developing and low-income countries where malnutrition and environmental exposures often co-occur. This study is not the first to find nutritional status important to mercury related immunomodulation [26,27]; however, it is provides additional evidence of the importance of adjusting for nutritional status and the first to observe that mercury related immunomodulation is antibody specific. Previously it was suggested that differences between the Faroe Island study, which observed a slight but insignificant negative association [25], and the NHANES study, where a negative association was found in nutrient sufficient children and a positive association in nutrient deficient boys [26,27], may relate to antibody type differences (live inactive vs. toxoid), controlling for nutritional status, or general population differences. Antibody differences were observed as subunit vaccines appeared to not be strongly associated by mercury as consistent associations were not observed with the two examined in this study (hepatitis B and Hib). Similar to the NHANES studies, by including an addition adjustment for nutritional status we observed an association between mercury and antibody titers; however, the direction of this association was different in this study. Here we observed that poor nutritional status (−1 HAZ) and near normal nutritional status (0 WHZ) were correlated with increased odds 7.3 and 4.3 increased odds of not being a responder against measles, respectively. Better nutritional status (+1 HAZ) was found to be protective. The differences between this study and the NHANES study may reflect differences in populations as children in this study region were on the whole were more nutrient deficient compared to their peers in the US. Our results in combination with other literature indicate that the impact of mercury on immune function is important but complex; further research is needed to elucidate the underlying mechanisms behind immunosuppression and stimulation related to antibody titers. 

Lastly, unrelated to mercury exposure or malnutrition, we observed that older children in native communities were 2.6 times more likely to not have antibodies levels protective against Hib, which has significant public health implications. This finding is consistent with studies that have found differences in Hib conjugate vaccine response in Native American populations [61,62] and increased risk of invasive Hib among Aboriginal populations in Australia [63,64] and Native American populations in the US and Canada [65,66]. The reduction observed in our study was only in older children (4 to <8 years), which may indicate that these children initially mount an immune response to Hib, but eventually lose protective levels. Factors that may explain the observed reduction in native-communities include vaccine quality, genetic susceptibility, and community differences in health and other resource access. We hypothesize that differences related to genetic susceptibility and disparities between native and non-native communities are more likely than poor vaccine quality because, nearly all young children (94%) residing in a native community had protective antibody levels against Hib. In addition, we did not observe a reduction in protection related to rural native communities with other vaccines, and in particular, those that are more sensitive to heat (MMR); therefore, we believe that it is unlikely that the observed antibody reduction was associated with decreased vaccine efficacy from cold chain failure [67].

The reduced protection observed in native communities could relate to genetic differences between ethnic groups, though we have limited knowledge of a child’s ethnic background and genetic factors were not measured in this study. Genetics have an important influence on variability in immune response and several single nucleotide polymorphisms, both inside and outside the major histocompatibility complex, have been associated with low antibody response to other vaccinations including hepatitis B [68] and antimalarial protection [69,70]. Concerning *H. influenzae*, heritability accounts for approximately 51% of variability in antibody response to Hib vaccination [71] and ethnic differences have been previously associated with increased disease incidence and reduced vaccine efficacy. Prior to *H. influenzae* type B vaccination and despite high vaccination coverage to this disease, Alaska Natives [72], American Indians [62,73], and indigenous populations in Australia [74] have greater disease rates and lower antibody levels. The variation in immune response in Navajo and Alaskan Eskimo populations may be in part related to genetics, as increased Hib disease susceptibility and antibody level were associated with loci interactions and a polymorphism in the A2 gene, a gene responsible for encoding light antibody chains [61,75]. Vaccine efficacy for such populations is problematic as the majority of conjugate vaccines to Hib assessed in the North American native populations have low efficacy in young children (<2 years) [62,72,76]. In a study that compared the immunogenicity of four Hib conjugate vaccines in Alaska Native infants, the antibody response to the different vaccines was large and only one vaccine induced a response with the first dose [76]. There is a need to better address ethnic differences in our population. Lastly, disparities in socioeconomic status or community access to health and other resources may relate to the reduced immune response; however, teasing out these differences from the native community variable was difficult. Clean water access was also observed to be correlated with decreased Hib titers, but the variation described by this variable was nearly indistinguishable from that of the native community variable. The lower antibody concentrations in these native communities warrants further examination to assess how an alternative vaccination schedule or different vaccine conjugate could improve vaccination response. 

## 5. Conclusions

This cross-sectional study identified that mercury exposure separately and in combination with nutritional status has the potential to influence child immune response in the ACR. Though the study was underpowered with a small sample size, we found it appropriate to conduct an exploratory analysis of child immune impacts related to mercury exposure, health status, and their interaction. The impact of mercury exposure on child antibodies was not unidirectional, as we observed correlations with increased and decreased antibodies. These antibody-dependent impacts indicate that the impact of mercury on immune function is complex and suggest that exposure could have disease specific outcomes. Immunomodulation from hair mercury and poor health interactions were observed for a few antibodies, but the association was particularly prominent with reduced measles titers, which indicates that exacerbated child health outcomes can occur in situations where mercury exposure and poor health status co-occur. Additionally, we observed that older children residing in native communities were nearly three times more likely to not have antibodies levels protective against Hib. This is an important finding for public health as it suggests that community differences, in terms of health access or socioeconomic status, and underlying ethnic group differences may be influencing seroconversion from vaccination against *H. influenzae*. Because these observations were made with a smaller than intended sample size and limited child vaccination history, the impacts and directionality observed in this population should be followed up with a prospective study that would have improved access to vaccination records and have the ability to measure vaccine response following vaccination and booster administration.

## Figures and Tables

**Figure 1 ijerph-16-00638-f001:**
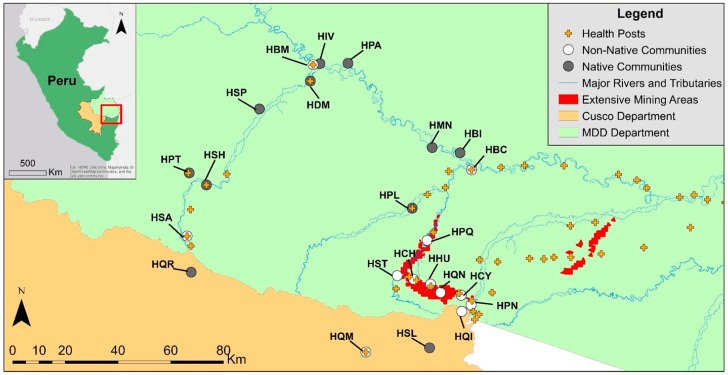
Map of study communities, including non-native (open circle) and native (filled circle) communities, in the Amarakaeri Communal Reserve in Peru shown in reference to extensive mining areas. Study communities include those upstream of active mining, HQR—Queros, HSA—Salvacion, HSH—Shintuya, HPT—Palotoa Teparo, HSP—Shipetiari, HDM—Diamante, HBM—Boca Manu, HIV—Isa de los Valles, and HPA—Puerto Azul, and communities near and downstream of mining activities, HMN—Masenawa, HBI—Boca Isiriwe, HBC—Boca Colorado, HPL—Puerto Luz, HPQ—Puquiri, HST—Setapo, HCH—Choque, HHU—Huepetuhue, HQN—Quebrada Nueva, HCY—Caychihue, HPN—Punquiri, HQI—Quimiri, HQM—Quincemil, and HSL—San Lorenzo.

**Figure 2 ijerph-16-00638-f002:**
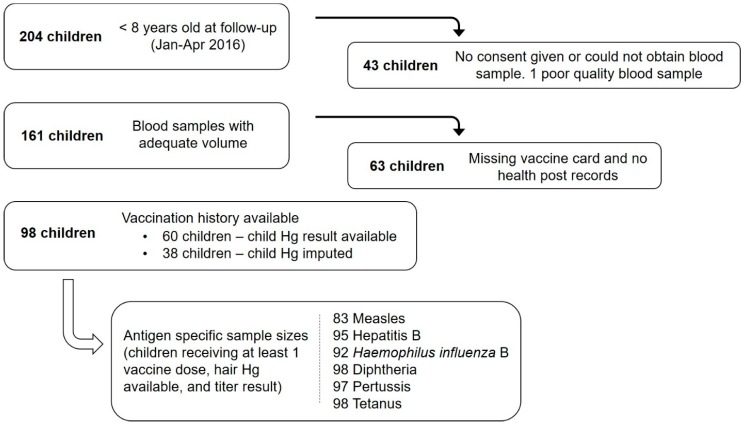
Flow diagram indicating the number of children (<8 years) that were contacted at follow-up, participated in follow-up, provided a blood sample, had baseline hair mercury data, and the number of samples ultimately included in the analysis.

**Figure 3 ijerph-16-00638-f003:**
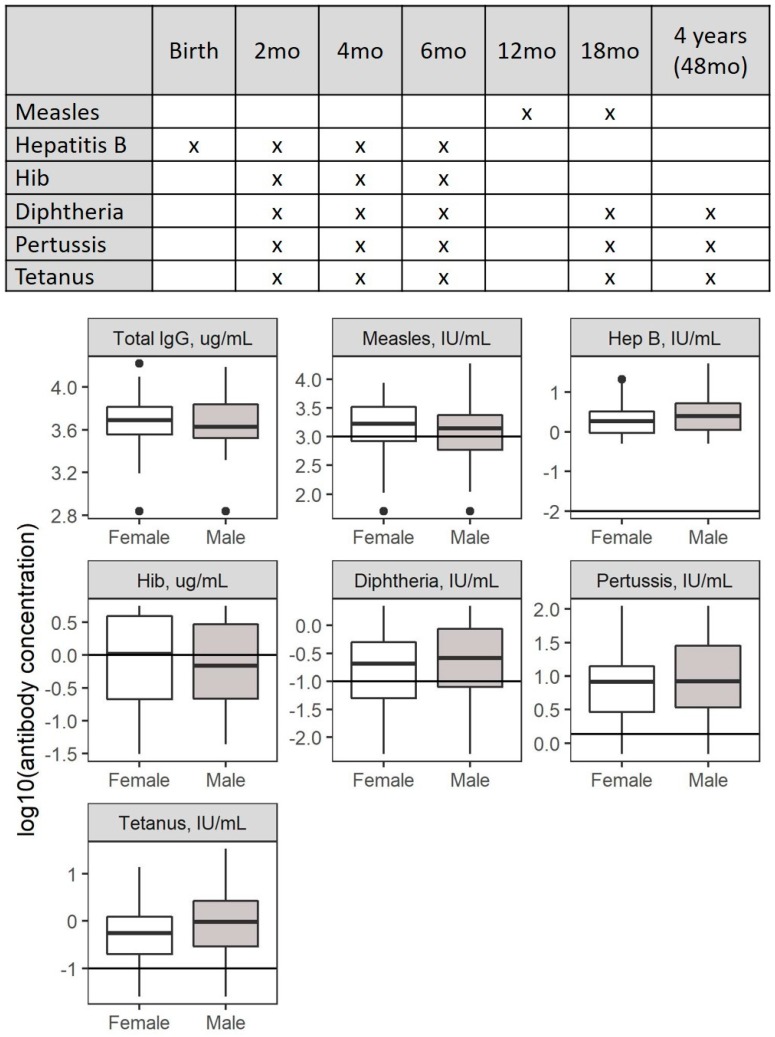
Peru’s vaccination schedule for study antibodies. Hep B (hepatitis B) is given initially as a shot at birth; Hep B, measles, diphtheria, pertussis, tetanus as Pentavalent vaccination at 2, 4, and 6 months; diphtheria, pertussis, and tetanus as DPT (diphtheria, pertussis, tetanus) at 18 months and 4 years; and measles as MMR (measles, mumps, rubella) at 12 and 18 months (top). Horizontal lines in boxplot figures denote antibody concentrations that denote protection (Hep B, diphtheria, tetanus) or responder threshold (measles, pertussis). mo: months.

**Figure 4 ijerph-16-00638-f004:**
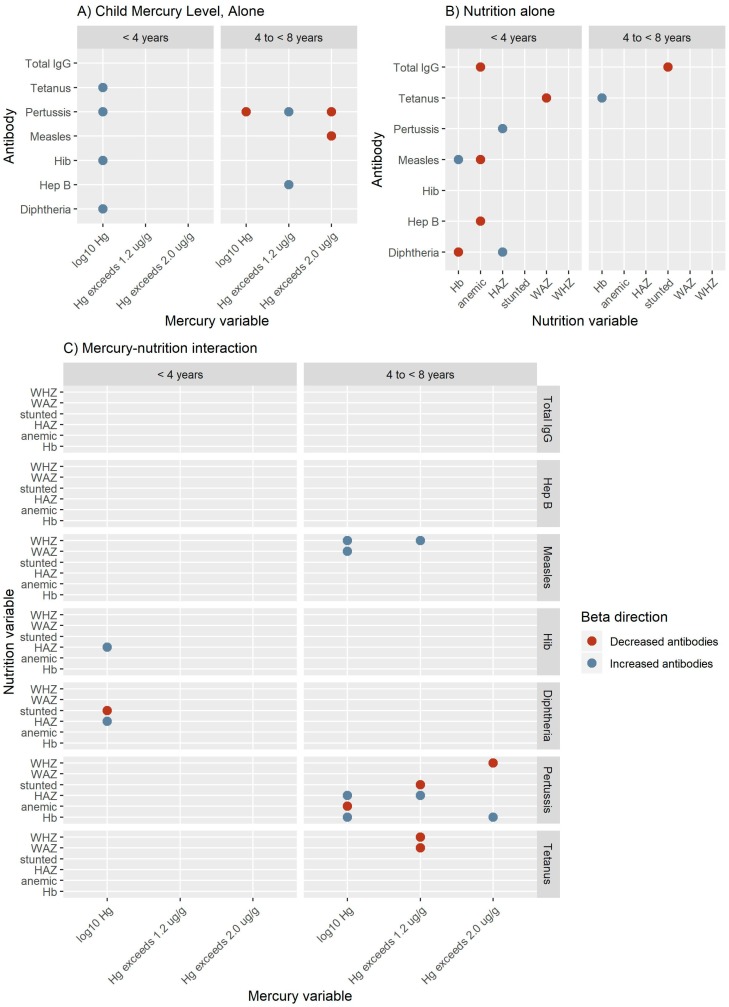
Summary of the mean beta direction from models estimating the effects of mercury (**A**), nutritional status (**B**), and mercury-nutritional status interaction (**C**) variables in continuous antibody models. In each section (A–C), the figures summarize beta estimates for the younger (<4 years) and older (4 to <8 years) children, in the left and right panel, respectively. Variables in all models included a mercury variable, a nutritional status variable, the interaction between mercury and the nutritional status variables, sex, and time (time since vaccination for vaccination antibodies or age for total IgG). Hib models for older children were also adjusted for native community. In the mercury alone panel (**A**) for a specific antibody and mercury variable, a dot signifies that that mercury variable was observed to be significant in at least one model when considered with each nutritional status variable. In the nutritional status alone panel (**B**) for a specific antibody and nutritional status variable, a dot signifies that that nutritional status variable was observed to be significant in at least one model when considered with each mercury variable. Dot color in A and B panels represents the sign of the average beta estimate for significant models. In the interaction panel (**C**), a dot signifies that the interaction between the indicated mercury and nutritional status variables was significant. Dot color in the C panel represents the sign of the interaction variable. SD: standard deviation; HAZ: height-for-age z-score; WAZ: weight-for-age z-score; WHZ: weight-for-height z-score; Hib: Haemophilus influenzae type B.

**Figure 5 ijerph-16-00638-f005:**
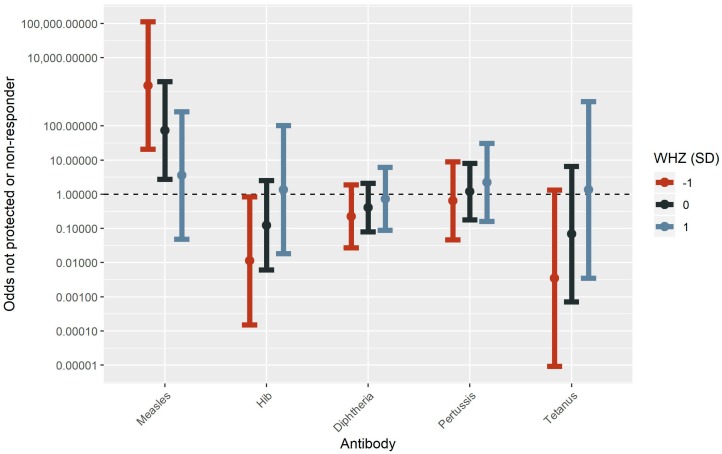
Odds ratios (±95% CI) for the interaction between child hair mercury exceeding 1.2 µg/g and WHZ score, displayed at three levels (−1 SD, 0 SD, and 1 SD), for having antibody concentrations that do not denote protection (Hep B, diphtheria, tetanus) or responder thresholds (measles, pertussis). Presented results are from older child (4 to <8 years) models that controlled for a mercury variable, a nutritional status variable, sex, and time (time since vaccination for vaccination antibodies or age for total IgG). Hib models were also adjusted for native community.

**Table 1 ijerph-16-00638-t001:** Characteristics of the Study Population by Community Type (native vs. non-native) and in comparison to a US Population using NHANES data.

Population Characteristic		Native Communities		Non-Native Communities		*p*-Value	All Study Children		NHANES		*p*-Value
		*N*	Mean (SD)	*N*	Mean (SD)		*N*	Mean (SD)	*N*	Mean (SD)	
Age, years											
	<4	22	2.5 (1.0)	46	2.6 (1.0)	0.978	68	2.6 (1.0)	239	2.4 (0.5)	0.184
	4 to <8	29	5.7 (1.1)	64	5.6 (0.9)	0.581	93	5.6 (1.0)	214	4.5 (0.5)	<0.001
Sex											
	Female	25	-	59	-	-	84	-	238	-	-
	Male	26	-	51	-	-	77	-	215	-	-
Child hair Hg											
	Hg content, µg/g	35	3.8 (2.6)	78	1.0 (2.7)	<0.001	113	1.5 (3.2)	453	0.2 (2.7)	<0.001
	Exceed 1.2 µg/g, %	88.6	-	42.3	-	-	56.6	-	2.9	-	-
	Exceed 2.0 µg/g, %	82.9	-	24.3	-	-	42.5	-	1.5	-	-
Parental average hair Hg											
	Hg content, µg/g	51	5.1 (2.1)	107	1.6 (2.4)	<0.001	158	2.3 (2.7)	-	-	-
Nutritional status											
	Hemoglobin, g/dL	50	10.9 (1.2)	108	11.6 (1.2)	0.001	158	11.3 (1.2)	453	12.6 (0.8)	<0.001
	Anemic, %	52	-	27.8	-	-	35.4	-	1.5	-	-
	Stunted, %	18.4	-	6.8	-	-	10.5	-	1.3	-	-
	HAZ	49	−1.3 (1.0)	103	−0.6 (1.0)	<0.001	152	−0.8 (1.0)	453	1.0 (1.4)	<0.001
	WAZ	50	−0.6 (1.1)	107	0.1 (1.2)	0.001	157	−0.1 (1.2)	453	0.9 (1.3)	<0.001
	WHZ	48	0.5 (1.0)	99	0.7 (1.1)	0.189	147	0.6 (1.1)	453	0.4 (1.1)	0.026
Vaccination Coverage, % ^a^											
	Measles	68.6	-	60.0	-	-	62.7	-	-	-	-
	Hepatitis B	74.5	-	70.0	-	-	71.4	-	-	-	-
	Hib	74.5	-	66.4	-	-	68.9	-	-	-	-
	Diphtheria	76.5	-	73.6	-	-	74.5	-	-	-	-
	Pertussis	76.5	-	72.7	-	-	73.9	-	-	-	-
	Tetanus	76.5	-	73.6	-	-	74.5	-	-	-	-
Vaccine Coverage, % ^b^											
	Hepatitis B (birth dose)	62.2	-	73.8	-	-	69.4	-	-	-	-
	Pentavalent	89.2	-	86.9	-	-	87.8	-	-	-	-
	DPT	64.9	-	63.9	-	-	64.3	-	-	-	-
	MMR	45.9	-	44.2	-	-	44.9	-	-	-	-
Antibody Titer ^c^											
	Total IgG (µg/mL)	51	4881.9 (1.8)	110	4464.0 (1.7)	0.181	161	4592.4 (1.7)	-	-	-
	Measles (IU/mL)	51	1034.6 (3.0)	110	1400.7 (3.4)	0.029	161	1272.5 (3.3)	-	-	-
	Hepatitis B (IU/mL)	51	2.0 (3.2)	110	2.1 (2.9)	0.943	161	2.1 (3.0)	-	-	-
	Hib (µg/mL)	51	0.4 (4.5)	110	1.0 (4.4)	0.004	161	0.8 (4.6)	-	-	-
	Diphtheria (IU/mL)	51	0.1 (5.8)	110	0.2 (5.4)	0.045	161	0.2 (5.6)	-	-	-
	Pertussis (IU/mL)	51	7.1 (4.3)	110	7.6 (3.9)	0.800	161	7.4 (4.0)	-	-	-
	Tetanus (IU/mL)	51	0.5 (4.5)	110	0.6 (4.8)	0.201	161	0.6 (4.7)	-	-	-

^a^ Expressed as a percent of children with receiving at least one booster for the respective antibody. ^b^ Expressed as a percent of children with vaccination history that is up-to-date according to the Peruvian Ministry of Health vaccination schedule. ^c^ Among children with at least one dose of the corresponding vaccine. NHANES: National Health and Nutrition Examination Survey; SD: standard deviation; HAZ: height-for-age z-score; WAZ: weight-for-age z-score; WHZ: weight-for-height z-score; Hib: Haemophilus influenzae type B; DPT: diphtheria, pertussis, tetanus; MMR: measles, mumps, rubella.

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
