# Peer review of "Mercury Exposure and Poor Nutritional Status Reduce Response to Six Expanded Program on Immunization Vaccines in Children: An Observational Cohort Study of Communities Affected by Gold Mining in the Peruvian Amazon"

_ijerph, 2019, doi:10.3390/ijerph16040638_

Reviewer 1 Report

The manuscript is very well written! The authors provided sufficient background information and I was able to easily follow the study design. I have several general comments:

1)The authors did not mention if the vaccines given to the children were preserved with thimerosal. In the US, thimerosal was taken out from many vaccines. Nevertheless, some of the multiple vaccines are still preserved with thimerosal. What is the case for Peru? 

2) It would have been nice for the authors to identify the children who were sampled during the baseline study and the follow-up study and undertake a separate analysis from the whole, randomized group. 

3) A clarification is needed as to what constitutes the native and non-native group. Listing the communities is not enough.  Are the authors referring to the indigenous tribes being the native group? What is the origin of the non-native group?  

4) The shift in antibodies response with age is puzzling, which raises the question if the authors can say with certainty that environmental conditions and mercury, in particular, affect the immune response of the study group. I suggest the authors approach this conclusion cautiously. For instance, in sentences like the one on line 561-562, use "has potential" instead of "influence".

Author Response

Please find our comments (highlighted in yellow) in the attached document. 

Reviewer 2 Report

This is a report analyzing mercury exposure and immunological reaction, and it is valuable because further consideration including nutritional status has been approached. This reviewer believes that such analysis including combined effects will bring important knowledge.

Firstly, the authors are requested to check whether the vaccine formulation does not contain mercury compounds such as Thimerosal. Thimerosal and its metabolite ethylmercury may affect the assessment of mercury exposure using hair samples.

Line 99-102: The authors reported that “Data are from an observational cohort study in Madre de Dios (MDD), Peru, a gold-mining region where the majority of the population have hair mercury content exceeding a level associated with impaired child development (1.2 μg/g) [31-33].” Then, they set this hair mercury level of 1.2 ppm as a cut-off point for the following analysis. However, this reviewer cannot identify the ground in the the references 31-33. Furthermore, they also used 2 ppm as another cut-off point. Please show the theoretical basis of these numbers. Similar questions are also raised in other places (Line 233-234). How do the authors convert the EPA RfD value from cord blood mercury content or maternal hair content to that of child hair?

Since the RfD had been determined based on the adverse effects on neurobehavioral changes, do the authors consider analyzing mercury values simply such as dividing it into quartiles?

Lines 75-77: In the sentence “Anemic conditions related to mercury exposure may occur from impaired hemoglobin function from mercury competing with iron for binding sites [14] and have been observed in the same region”, what have been observed in the “same region” ?

2.3. Hair mercury analysis: Since inhalation exposure to mercury vapor is possible (Line 120-121), did the authors wash the hair samples prior to Hg determination to exclude the possibility of external contamination of hair materials with mercury vapor? This is the same in the mercury determination using toenail samples.

Do the authors use the Hg data from toenail samples?

Lines 249-250 & Figure 3: Since antibody concentrations are log-transformed, the figures indicating the antibody concentrations should be shown logarithmically.

In Figure 4(A), The results are complicated and they seem to be inconsistent sometimes. For example, mercury exposure exceeding 1.2 ppm increased the antibody concentration of Pertussis, and that exceeding 2.0 ppm decreased the concentration. Although the authors showed these phenomenon in Discussion, what mechanism do the authors think about?

If the authors integrate the factors of malnutrition into one index, it is easy to understand.

Line 380: The meaning of the sentence “Associations with parental hair mercury were also observed.” is unclear.

Figure 5: The explanation of the vertical axis of the graph is incorrect; “non-reponder” will be non-responder.

Author Response

(The authors gave the same response as above.)

Reviewer 3 Report

I think the article is very interesting and brings new information on mercury effects related to vaccines. Even though it is preliminary data it is important to publish this kind of data to add a better understanding to other effects of mercury, mainly when affect such an important public health matter such as respond to immunization programs.

I suggest some minors’ changes, that are specified  below.

Lines 123-126

I strongly suggest mentioning the Peruvian anemia prevalent for both WCBA and children under 5. To situate the reader on the country reality how the MDD communities are in comparison with the country situation.

Nail elements analysis. (lines: 184-197)

Is this data presented in any part of this study? I see the models used mercury in hair but not in the toenail. Is there any of the elements mentioned in this paragraph used in this article? Why is this methodology presented? I suggest removing it or use the data. It will be interested in see if there is any interaction with the essential elements and the toxic ones. Is suggested in the literature there is an antagonist relationship between Hg-Se.

Lines (235-236)

I don’t think US population is the best option, especially when referring to weight and height conditions, where race difference could influence some of the measurements. Isn't there any data from the Peruvian Health Ministry from the whole country? Is there any regional data that can be accessed? The Brazilian Health system has this kind of data and is more comparable with this population... I strongly suggest the author include comparison with more alike populations too.

Lines (261-262)

Were there any cases in which the children were more than 4 years old and did a partial vaccination? (means that didn’t apply all the doses suggested) If so, how this was taken into account in the model?

Lines (263-265)

I understand that time since last vaccination affects the respond of antigens. What about the number of the dose of vaccination, is that could affect the antigen response? If so, how this is taken into account in the model if it is 1st, 2nd or 3rd dose?

 Author Response

Please find our comments (highlighted in yellow) in the attached document. 

Reviewer 4 Report

Very nice study and paper. Please see the attached for some minor considerations.

Author Response

Please find our comments (highlighted in yellow) in the attached document. 

Round  2

Reviewer 2 Report

This reviewer has no additional comments.